# Fabrication and Characterization of Modified Graphene Oxide/PAN Hybrid Nanofiber Membrane

**DOI:** 10.3390/membranes9090122

**Published:** 2019-09-19

**Authors:** Jian Hou, Jaehan Yun, Hongsik Byun

**Affiliations:** 1Department of Chemical Engineering, Zibo Vocational Institute, Zibo 255314, China; houjimmy@naver.com; 2Department of Chemical Engineering, Keimyung University, Daegu 704701, Korea; ruri7220@naver.com

**Keywords:** modified GO, PAN, hybrid nanofiber membrane, polydiallyldimethylammonium chloride (PDDA)

## Abstract

In this study, a series of novel modified graphene oxide (MGO)/polyacrylonitrile (PAN) hybrid nanofiber membranes were fabricated by electrospinning a PAN solution containing up to 1.0 wt.% MGO. The GO was initially prepared by a time-saving improved Hummer’s method. Subsequently, the formation of GO was confirmed by scanning electron microscopy (SEM), AFM, Fourier-transform infrared spectroscopy (FT–IR), and Raman spectroscopy. This study also prepared the modified GO with polydiallyldimethylammonium chloride (GP) by using a simple surface post-treatment method to improve its dispersion. Varying amounts of GP were incorporated into PAN nanofibers for the better properties of GP/PAN nanofibers, such as hydrophilicity, mechanical properties, and so on. The resulting GP/PAN hybrid nanofiber membranes were characterized by SEM, FTIR, contact angle, and thermal and mechanical properties. These results showed that the hydrophilic and mechanical properties of GP/PAN hybrid nanofiber membranes were dramatically improved, i.e., 50% improvement for hydrophilicity and 3–4 times higher strength for mechanical property, which indicated the possibility for water treatment application. In addition, the notably improved thermal stability results showed that the hybrid nanofiber membranes could also be a potential candidate for the secondary battery separator.

## 1. Introduction

Nanofibers, which are important nanomaterials, have attracted increasing attention over the last two decades. A nanofiber generally refers to a fiber having a diameter less than 100 nm. However, fibers with diameters less than 1000 nm (micro-) can also be defined as nanofibers in the fiber industry field [1,2]. Many researches on the fabrication and application of nanofibers have attracted great interest from both academia and industry. The general fabricating methods of nanofibers are electrospinning, self-assembly, phase separation, melting blown, and nanoporous template. Among these methods, electrospinning is the most promising method for producing continuous nanofibers on a large scale, and the fiber diameter can be easily adjusted. Also, electrospinning is a relatively simple and fast process to produce nanofibers [3,4,5,6]. A number of organic polymers, including natural and biomaterial polymers, have been successfully spun into nanofibers using electrospinning. Electrospun nanofibers have many applications due to their excellent properties, such as very small pores, large surface area, and higher porosity compared with regular fibers, which are produced by conventional melt and wet spinning processes [7,8,9]. Until now, nanofibers have been widely reported and applied in various fields, such as filtrations, wound dressings, tissue engineering scaffolds, protective clothing, electronics, sensors, energy storage, and drug delivery materials. Other new applications have been continuously explored for these nanofibers [10,11,12,13].

Recent researches have shown that polyacrylonitrile (PAN) was generally chosen to prepare nanofiber membranes and used as a water treatment membrane or Li-ion battery (LIB) separator [14,15]. Firstly, PAN is one of the most extensively used polymers in electrospinning because of its excellent properties [16,17]. Furthermore, electrospun PAN nanofibers possess a variety of advantages, including thermal stability, resistance to most solvents, high strength, and so on [18,19]. In addition, in order to further improve the hydrophilicity and other properties of PAN nanofibers, some nanofillers such as nanosized TiO_2_, ZnO, graphene oxide (GO), etc. have been added to the polymer solution to prepare hybrid nanofibers [20,21,22]. Instead, GO contains several oxygen-containing groups, such as epoxy and hydroxyl groups at basal plane and carbonyl and carboxylic groups at the edge. As a result, GO has higher amphiphilicity and reactivity than other nano-materials, thus making it a more suitable nanofiller for improving the properties of hybrid nanofiber membranes than other nanomaterials [23,24,25]. Recently, it has been found that adding GO nanofillers can greatly enhance the mechanical properties of hybrid nanofiber membranes. However, due to the limitation of dispersibility in the polymer precursor solutions, uniform distribution of excess GO nanofillers over 0.4 wt.% into nanofibers is still challenging [26]. Therefore, improving the dispersibility of GO in polymer precursor solution has become a hot topic in recent research [25,27,28,29].

In this report, GO was firstly prepared from flake graphite using an improved Hummer’s method. We also modified the surface of GO with a simple method in order to improve its dispersibility. This modified GO was mixed with a PAN polymer solution to form a homogeneous precursor solution. The final precursor solution was then fabricated as the hybrid nanofiber membranes via the electrospinning process. The resulting nanofiber membranes were characterized by scanning electron microscopy (SEM), Fourier-transform infrared spectroscopy (FT–IR), contact angle, thermogravimetric analysis (TGA), and mechanical property.

## 2. Materials and Methods

### 2.1. Materials

Graphite flake (Bay Carbon Inc., Baycity, MI, USA), sodium nitrate (NaNO_3_, ≥ 99%, Sigma–Aldrich Korea, Seoul, Korea), KMnO_4_ (99%, Sigma–Aldrich), sulfuric acid (98%, Duksan, Seoul, Korea), and hydrogen peroxide (H_2_O_2_, 35%, Samchen Co., Ltd., Seoul, Korea) were used for the synthesis of GO. Materials used to manufacture the nanofibers were polyacrylonitrile (PAN, Mw 150,000, Sigma–Aldrich) and N,N–dimethyl formamide (DMF, 99.5%, Duksan). Polydiallyldimethylammonium chloride (PDDA) was used as a surfactant to modify the surface of GO. All chemicals were used without further purification.

### 2.2. Synthesis of Graphene Oxide (GO) and Modified GO by Polydiallyldimethylammonium (PDDA) (GP)

GO was synthesized from graphite using a time-saving improved Hummer’s method. Briefly, graphite flake (2 g), NaNO_3_ (1.52 g), and sulfuric acid (68 mL) were mixed and stirred for 10 min, and then 9 g of KMnO_4_ was slowly added into the mixture. The mixture was then allowed to heat to 60 °C and stirred for 40 min. One hundred milliliters of distilled water (DI water) was added to the solution, followed by mixing for 1 h at 90 °C [30]. The temperature of the mixture was decreased to 50 °C, followed by the addition of 100 mL DI water again. The mixture was then fully dispersed by sonication for 3 h and then the mixture was filtered using a filter paper (No. 2, Advantec, Dublin, CA, USA). In order to remove the unreacted residue in the filtrate, DI water (180 mL), H_2_O_2_ (6 mL), and sulfuric acid (1 mL) were added, respectively, and then the mixed filtrate was centrifuged at 4000 rpm for 30 min [31]. The filtrate was washed several times with DI water and ethanol, then thoroughly dried in a vacuum oven for 24 h to obtain the GO product. Dried GO (0.5 g) was fully dispersed in water (100 mL) for 1 h by sonication and centrifuged for 30 min. Then, the final GO was obtained by filtering the GO solution and dried in a vacuum oven. The fully dried GO (0.05 g) was dispersed in DI water (99.95 g) to prepare GO suspension for modification with the help of sonication. PDDA (0.033 g) was added to the GO suspension and stirred for 24 h. Finally, the GO + PDDA (GP) was obtained by filtering and vacuum drying.

### 2.3. Fabrication of Polyacrylonitrile (PAN) and Graphene Oxide Modified by Polydiallyldimethylammonium Chloride/Polyacrylonitrile (GP/PAN) Hybrid Nanofiber Membranes via Electrospinning

The PAN nanofiber membranes incorporated with varying amounts of GP were fabricated via the electrospinning method. A homogenous precursor solution of PAN and GP listed in Table 1 was prepared and a series of GP (0–1.0 wt.%) was initially dispersed in DMF by sonication for 1 h. PAN powder was then added to the above solution and fully dissolved by stirring for 24 h at 50 °C. Then, the prepared solutions were filled into a 5 mL syringe with a 23-gauge needle. The syringe was positioned vertically for 30 min or more to remove the air bubbles. Subsequently, the syringe was installed in an automated syringe pump (KDS100, KD Scientific Korea Inc., Seoul, Korea) to control the ejection speed of the solution, and the voltage supply equipment used was a CPS 60K02VIT (CHUNGPA EMT co., Ltd., Seoul, Korea). Prepared spinning solutions were electrospun on to aluminum foil under an electrical field with optimized voltage (15 kV), flow rate (0.8 mL/h), distance between the needle tip and the collector (15 cm), relative humidity (20–40%) and room temperature.

### 2.4. Characterization of Synthesized Graphene Oxide GO and Graphene Oxide Modified by Polydiallyldimethylammonium Chloride/Polyacrylonitrile (GP/P) Hybrid Nanofiber Membranes

The morphological analysis of the GO was carried out by FESEM (Field Emission Scanning Electron Microscope, Hitachi Korea, S-4200, Seoul, Korea). An AFM (Atomic Force Microscope, NanoSurf Korea, Seoul, Korea) was used to analyze the surface morphology of GO. FT–IR (JASCO, FT/IR-620, Tokyo, Japan) was used to observe the various functional groups of synthesized GO. The Raman test was performed on Thermo DXR Raman Spectroscopy (Thermo Fisher Scientific Korea, Seoul, Korea) with a 532 nm laser and 1–5 mW power.

The optical image of nanofibers was taken with a Pentax K-S2 DSLR Camera and the structural features of nanofibers were examined by scanning electron microscope (SEM, JSM5410, Tokyo, Japan) after coating with a gold target. The pore diameters of nanofiber membranes were analyzed with a capillary porometer (Porolux 1000, IB-FT GmbH, Berlin, German) under wet and dry conditions using a Porewick standard solution, and the effective diameter of the nanofibers was fixed at 1.9 cm. The porosity of the samples (5 cm × 5 cm) was examined by measuring the dry and wet weights of the nanofiber membranes after soaking in n–butanol for 1 h. The hydrophilicity of nanofiber membranes was confirmed with a contact angle analyzer (Phoenix 300, SEO Inc, Seoul, Korea) using a water droplet. The mechanical testing was performed using a universal tensometer (MYUNGJI TECH, TENSO, Seoul, Korea) with a crosshead speed of 500 mm/min and a sample size of 100 mm × 30 mm.

## 3. Results

### 3.1. Characterization of Synthesized Graphene Oxide (GO)

The SEM image (Figure 1a) indicates that GO is few micro-meters in length and width. Figure 1b shows the AFM image of GO, and the line scan conducted on this sample. Figure 1c shows the height profile of the line scan. The thickness of the synthesized GO is approximately 0.9–1 nm. It is reported that the thickness of perfect single layer of GO was about 0.8 nm. Therefore, we can infer that this synthesized GO is almost a single layer.

FT–IR spectra is recognized as an important tool for characterization of functional group in GO, so FT–IR test was performed and the spectra is shown in Figure 2. Among the salient features observed are the bands at 1740 cm^−1^, 1630 cm^−1^, 1226 cm^−1^, and 1040 cm^−1^, corresponding to the stretching modes of C=O, –C=C, C–O, and C–O–C groups in GO, respectively [32,33].

As shown in Figure 3, the Raman spectrum of synthesized GO displays a D-band at 1353 cm^−1^ and a broad G-band at 1587 cm^−1^, which were the typical spectra of GO [34]. Ramakrishnan et al. reported that the G-band and D-band represented different carbon networks. The G-band was due to the characteristics of all sp2-hybridized carbon networks, resulting in the first-order scattering in the Brillouin zone center, while the D-band, caused by the attachment of oxygen groups on the carbon basal plane, showed the structural imperfections [35].

### 3.2. Characterization of Graphene Oxide Modified by Polydiallyldimethylammonium Chloride/Polyacrylonitrile (GP/PAN) Hybrid Nanofiber Membranes

Figure 4 and Figure 5 show the digital photos and the SEM images of representative PAN and a series of GP/PAN hybrid nanofiber membranes, respectively. It can be easily observed that the more GP content in the PAN, the darker the color that was obtained [30]. Based on our previous research, the loading amount of bare GO into PAN nanofiber membranes by electrospinning generally reached up to 0.4 wt.% with respect to the polymer concentration, due to the limited dispersity of GO in a polymer precursor solution [26]. Only through the process of simple surface modification, the modified GO (GP) nanofiller was reliably added up to 1.0 wt.% under the same electrospinning conditions. This modification involved the electrostatic interactions between PDDA and GO, which greatly improved the dispersity of GO in polymer precursor solution as well as reliable electrospinning of nanofibers without a needle blocking problem. Meanwhile, as the amount of GP increased, the average diameter of nanofibers increased from 0.563 µm to 0.618 µm. This is reasonable, since as the amount of GP increases in the polymer solution, the viscosity of the final precursor solution increases due to the low amount of solvent, resulting in thicker nanofibers. However, we think that this can only happen if GP is dispersed homogeneously in the polymer solution. Otherwise, the viscosity of the GP polymer solution will not increase consistently. The pore size and thickness of the nanofiber membranes were examined (Table 2). The PAN nanofiber membrane possessed an average pore size of 213.3 nm and a thickness of 77 µm. The hybrid nanofiber membranes gradually increased the pore size and the thickness with the increase of the GP content. These results demonstrate our previous research results once more, i.e., as the diameter of nanofibers increases, the pore size and thickness also increase [26].

To confirm the successful composite of GP into the nanofibers, the FT–IR of GP/PAN nanofibers were measured (Figure 6). It was easily recognized that GP was successfully added, due to the stronger specific GO functional group peaks (3200 cm^−1^ (OH peak), 1740 cm^−1^ (C=O), 1630 cm^−1^ (C=C), 1226 cm^−1^ (C-O), 1040 cm^−1^ (C–O–C)) in the FT–IR spectra [36,37]. Also, in Figure 7, the contact angle results display an obvious decrease trend due to the various hydrophilic functional groups in the GO, which have also been checked in the FT–IR and some earlier published work [36,38].

The thermal stability of the PAN nanofiber membrane and GP/PAN hybrid nanofiber membranes is shown in Figure 8. Only two samples were measured. Under pyrolytic conditions, the PAN nanofiber membrane with and without nanofillers degraded in two steps. Compared with the PAN nanofiber membrane, the onset thermal decomposition temperature was increased from 300 °C to 340 °C. It could thus be concluded that with the addition of GP, the thermal stability of the nanofiber membrane increased. The noticeable improvement of the thermal stability could be contributed to both the original property of GO and the surface modified with PDDA.

Figure 9 shows the stress-strain curves of the PAN nanofiber membrane and series of the GP/PAN hybrid nanofiber membranes. Overall, the tensile strength of all hybrid nanofiber membranes had obviously improved, compared to the PAN nanofiber membrane. However, when the GP content increased to > 0.4 wt.%, the hybrid nanofiber membrane appeared to have a little lower stress, due to the reduction of the dispersion. When the GP content is more than 0.4 wt.%, it will be difficult to disperse uniformly in the PAN precursor solutions. Consequently, the beaded structures will be formed in the hybrid nanofiber membrane, which can also be confirmed in the SEM images of Figure 5f [26,37].

## 4. Conclusions

In this work, we focused on the improvement of GO’s dispersity in a PAN precursor solution for electrospinning and the fabrication of GP/PAN (GP: GO modified by PDDA) hybrid nanofiber membranes. To the best of our knowledge, according to the simple surface modification, the GO content in the polymer precursor solution can be added more than 0.4 wt.% (up to 1.0 wt.%) under the same electrospinning conditions for the first time. The prepared GO nanomaterials and hybrid nanofiber membranes were characterized by SEM, FT–IR, Raman spectra, contact angle, thermal and physical properties, etc. These results showed that the pore size and the thickness can be easily controlled by the content of GP. According to contact angle analysis, these fabricated PAN/GP hybrid nanofiber membranes exhibited better hydrophilic characteristics than the PAN nanofiber membrane. Lastly, the incorporation of the GP into the PAN nanofiber membranes generally improved their thermal stability and mechanical properties, compared to the PAN nanofiber membrane. Based on all the above results and the original advantage of nanofibers, the PAN/GP hybrid nanofiber membranes could be a promising candidate for water treatment membranes. Further research about water purification and battery performance is being investigated.

## Figures and Tables

**Figure 1 membranes-09-00122-f001:**
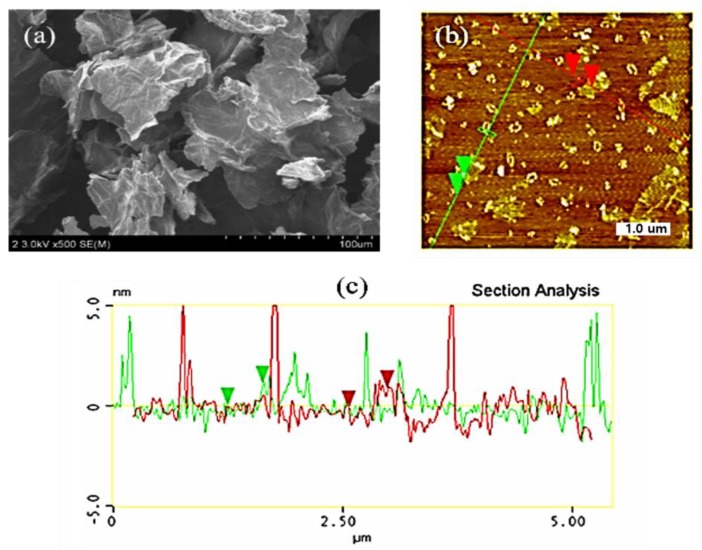
(**a**) field emission scanning electron microscopy (FESEM) image of graphene oxide (GO), (**b**) atomic force microscopy (AFM) image of GO, and (**c**) height profile of GO.

**Figure 2 membranes-09-00122-f002:**
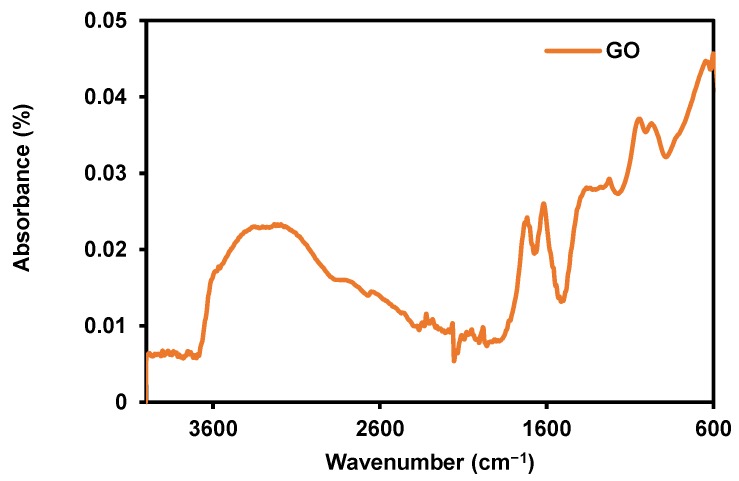
Fourier-transform infrared spectroscopy (FT–IR) of synthesized graphene oxide (GO).

**Figure 3 membranes-09-00122-f003:**
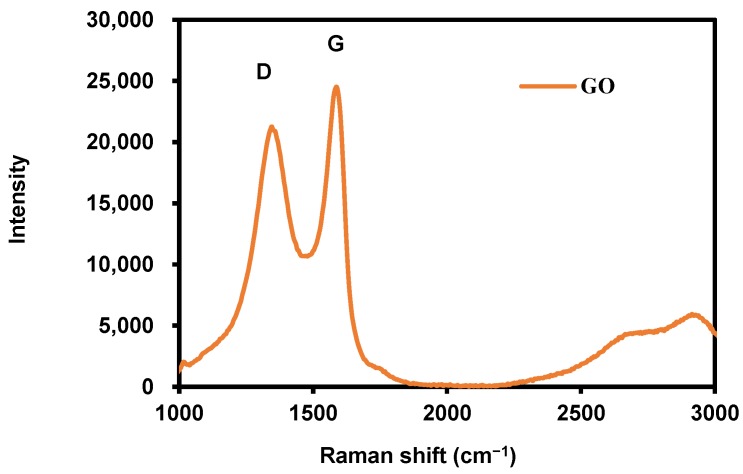
Raman spectra of synthesized graphene oxide (GO).

**Figure 4 membranes-09-00122-f004:**
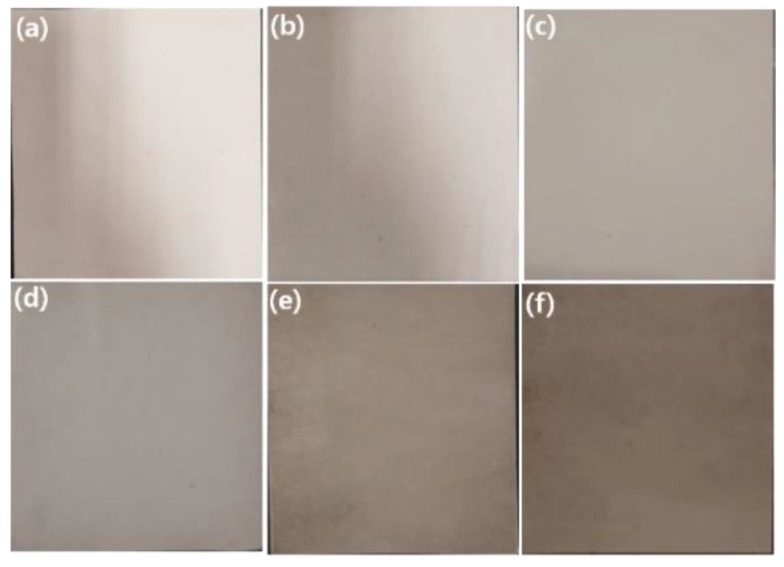
Digital photos of polyacrylonitrile (PAN), graphene oxide modified by polydiallyldimethylammonium chloride/polyacrylonitrile (GP/PAN) hybrid nanofiber membranes; (**a**) PAN nanofiber membrane, (**b**) GP01PAN, (**c**) GP02PAN, (**d**) GP03PAN, (**e**) GP04PAN, (**f**) GP10PAN.

**Figure 5 membranes-09-00122-f005:**
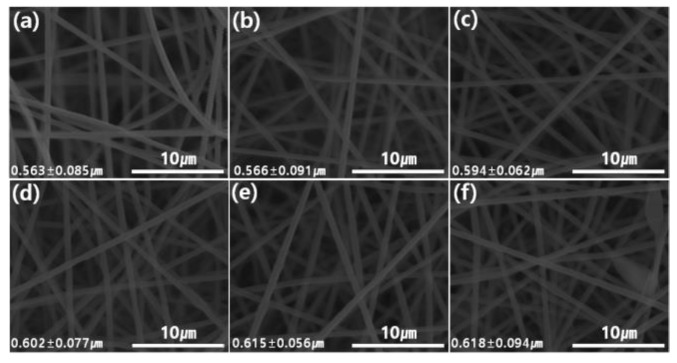
Scanning electron microscopy (SEM) images of polyacrylonitrile (PAN), graphene oxide modified by polydiallyldimethylammonium chloride/polyacrylonitrile (GP/PAN) hybrid nanofiber membranes; (**a**) PAN nanofiber membrane, (**b**) GP01PAN, (**c**) GP02PAN, (**d**) GP03PAN, (**e**) GP04PAN, (**f**) GP10PAN.

**Figure 6 membranes-09-00122-f006:**
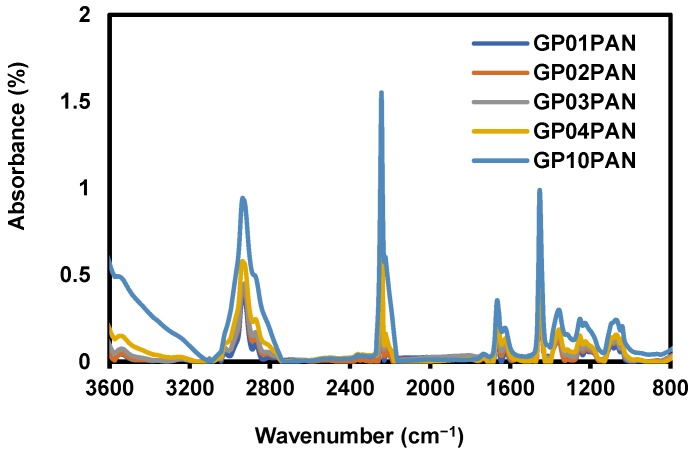
Fourier-transform infrared spectroscopy (FT–IR) spectra of graphene oxide modified by polydiallyldimethylammonium chloride/polyacrylonitrile (GP/PAN) hybrid nanofiber membranes.

**Figure 7 membranes-09-00122-f007:**
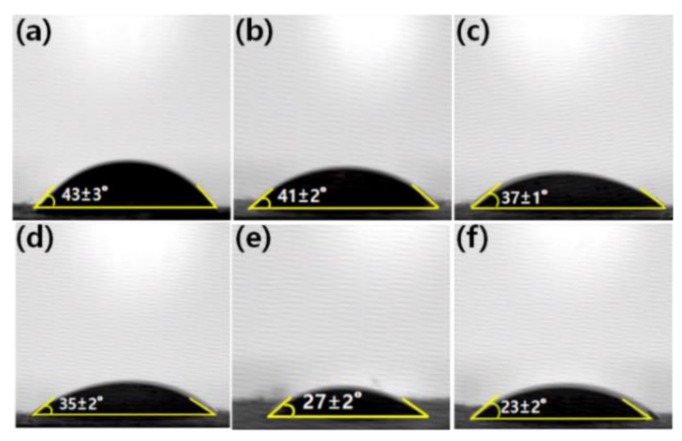
Contact angle of polyacrylonitrile (PAN), graphene oxide modified by polydiallyldimethylammonium chloride/polyacrylonitrile (GP/PAN) hybrid nanofiber membranes; (**a**) PAN nanofiber membrane, (**b**) GP01PAN, (**c**) GP02PAN, (**d**) GP03PAN, (**e**) GP04PAN, (**f**) GP10PAN.

**Figure 8 membranes-09-00122-f008:**
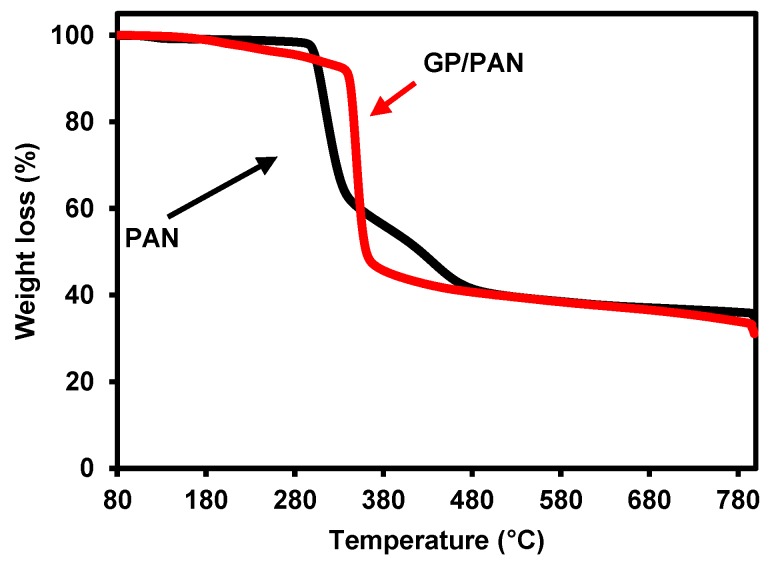
Thermogravimetric curves of polyacrylonitrile (PAN) and graphene oxide modified by polydiallyldimethylammonium chloride/polyacrylonitrile (GP10/PAN) hybrid nanofiber membranes.

**Figure 9 membranes-09-00122-f009:**
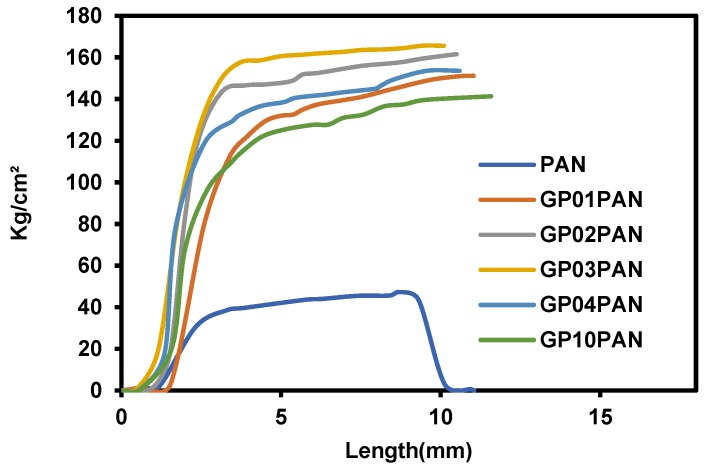
Tensile strength of polyacrylonitrile (PAN) nanofiber and graphene oxide modified by polydiallyldimethylammonium chloride/polyacrylonitrile (GP/PAN) hybrid nanofibers.

**Table 1 membranes-09-00122-t001:** Component ratio of polyacrylonitrile (PAN) and graphene oxide modified by polydiallyldimethylammonium chloride/polyacrylonitrile (GP/PAN) electrospinning solution. (DMF = N,N-dimethyl formamide).

Sample Name	GP (wt.%)	PAN (wt.%)	DMF (wt.%)
PAN	0	10	90.0
GP01PAN	0.1	10	89.9
GP02PAN	0.2	10	89.8
GP03PAN	0.3	10	89.7
GP04PAN	0.4	10	89.6
GP10PAN	1.0	10	89.0

**Table 2 membranes-09-00122-t002:** Pore size and thickness of polyacrylonitrile (PAN), graphene oxide modified by polydiallyldimethylammonium chloride/polyacrylonitrile (GP/PAN) nanofiber membranes.

Sample Name	Biggest Pore Size (nm)	Smallest Pore Size (nm)	Average Pore Size (nm)	Thickness (µm)
PAN	459.5	169.5	213.3	77 ± 3
GP01PAN	474.2	171.2	226.3	79 ± 2
GP02PAN	484.5	175.5	232.7	81 ± 2
GP03PAN	473.3	183.2	243.5	83 ± 3
GP04PAN	482.2	194.3	245.5	84 ± 2
GP10PAN	485.5	243.6	269.3	88 ± 3

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
