# Peer review of "Fabrication and Characterization of Modified Graphene Oxide/PAN Hybrid Nanofiber Membrane"

_membranes, 2019, doi:10.3390/membranes9090122_

Round 1
Reviewer 1 Report
This manuscript is dedicated to the fabrication and characterization of modified graphene oxide/PAN hybrid nanofiber membrane. The study employs the appropriate methodology which is accordingly discussed and explained.
This research is timely (Hybrid nanofiber membranes are of much growing interest) and quite valuable, and the authors addressed the topic in an original and at the same time easy to perceive way.
There are well-presented figures, as well as concise discussion which makes the manuscript an informative read.
There are some minor aspects of this good manuscript that need revision; thus, it is acceptable for publication after relatively minor revision:
1: Introduction is written well. The processes that concern bonding and structure especially concerning graphene and graphene oxide are also subject of theoretical modelling accessible by approaches such as Synthetic Growth Concept, e.g.,
Phys. Chem. C 2014, 118, 6514−6521Citing such works will enlarge the context of the present introduction.
2: Related to the Raman spectra: more discussion is necessary in relation to the D peak and its relation to significant structural deviations from sp2 bonded carbon.
3: The English of the manuscript is good, but it still needs some improvement, please, apply a spellchecker, and a stylistic review of the longer sentences.
4: Conclusions are concisely presenting the main results and their context which is excellent.
However, what is partially missing (and it will improve the message of the paper) is more concrete details (mentioning no more than 2-3 of the values obtained) thus giving better concreteness to the final message of the manuscript.
Author Response
Fabrication and characterization of modified graphene oxide/PAN hybrid nanofiber membrane
Response to comments by Reviewers:
We greatly thank all Reviewers for their constructive feedback of our work and for the helpful comments.
Reviewer #1:
Comments: This manuscript is dedicated to the fabrication and characterization of modified graphene oxide/PAN hybrid nanofiber membrane. The study employs the appropriate methodology which is accordingly discussed and explained.
This research is timely (Hybrid nanofiber membranes are of much growing interest) and quite valuable, and the authors addressed the topic in an original and at the same time easy to perceive way.
There are well-presented figures, as well as concise discussion which makes the manuscript an informative read.
There are some minor aspects of this good manuscript that need revision; thus, it is acceptable for publication after relatively minor revision:
COMMENTS TO THE AUTHOR(S):
1: Introduction is written well. The processes that concern bonding and structure especially concerning graphene and graphene oxide are also subject of theoretical modelling accessible by approaches such as Synthetic Growth Concept, e.g., Phys. Chem. C 2014, 118, 6514−6521. Citing such works will enlarge the context of the present introduction.
>> Thank you for your valuable comments. As your comment, we have read and cited such works which you mentioned e.g. Phys. Chem. C 2014, 118, 6514−6521
Goyenola, C.; Stafström, S.; Schmidt, S.; Hultman, L.; Gueorguiev, G. K. Carbon Fluoride, CFx: Structural Diversity as Predicted by First Principles. Phys. Chem. C 2014, 118(12), 6514–6521.2: Related to the Raman spectra: more discussion is necessary in relation to the D peak and its relation to significant structural deviations from sp2 bonded carbon.
>> Thank you for comments and sorry for the unclear explanation. However, since this explanation was based on the theory suggested by the Ramakrishnan et.al. we revised the sentence as below, showing the source of this explanation clearly.
Ramakrishnan et. al. reported that the G-band and D peak represented different carbon networks. The G-band was due to the characteristics of all sp2-hybridized carbon networks, resulting in the first-order scattering in the Brillouin zone center, while the D peak caused by the attachment of oxygen groups on the carbon basal plane showed the structural imperfections [35].
3: The English of the manuscript is good, but it still needs some improvement, please, apply a spellchecker, and a stylistic review of the longer sentences.
>> Thank you for your suggestion. As you suggested, we have used a spellchecker and grammarchecker to check this manuscript again and made some improvement.
4: Conclusions are concisely presenting the main results and their context which is excellent. However, what is partially missing (and it will improve the message of the paper) is more concrete details (mentioning no more than 2-3 of the values obtained) thus giving better concreteness to the final message of the manuscript.
>> Thanks for all the valuable comments. This suggestion will be a great help for us to prepare the next manuscript.
Reviewer 2 Report
General comments
The reviewed manuscript concerns preparation and analysis of Graphene Oxide dispersity in PAN nanofibrous matrix. Introduction of GO filler inside nanofibers could improve several materials’ properties including mechanical one. The effect of nanofillers’ presence inside nanofibers and modifications with PDDA was studied with several methods including FT-IR, SEM, TGA and Raman. The manuscript contains many typos and mistakes but don’t need language corrections. In such form, the manuscript cannot be published.
Specific comments
Line 10 were instead of was
Line 19 Statistical analysis would be appreciated
Line 48 „Especially, GO exists…” use instead GO contain several…
Line 56 Please cite work done by Pierini et al. Titled Electrospun poly(3-hexylthiophene)/poly(ethylene oxide)/graphene oxide composite nanofibers: effects of graphene oxide reduction, where they also modified GO to increase dispersity in nanofibers.
Line 57-60 The sentence is so long that it is impossible to understand it.
Line 77 Please remove a dot after 3 h.
Line 125 Figure 1. Please increase size of the pictures so that it fits the width of the text. Also, please include scales for AFM picture (X-Y) and height.
Line 131 Figure 2. Please increase the scale so that it is better visible where the peaks are. Also, for C=O and C=C there are single peaks not the range.
Line 151 Please do not name polymer solution a precursor solution because it is misleading.
Line 155-156 The most important part. Since you are trying to increase the GO content in Pan fibres, and take care of its homogenous dispersion, does the analysis you perform allow you to say that it is homogenously dispersed? What about TEM analysis?
Line 159 Since the fibres are quite large based on the SEM pictures you provide, what pore size did you measure? The pores between the fibres should be in micrometres, not in nanometres. Or maybe you mention the porosity of single nanofiber?
Line 164 The Figure 4 is useless, the info you gave in the text about darker colour of the material is enough.
Line 167 SEM images should be increased in size, and since you used FE SEM you could provide more detail of single nanofibers.
Line 188 Figure 8 Why the weight loss above 480 oC of nanofibrous material containing GO is higher comparing with PAN? It contains additionally GO, so it should be lower.
Line 198 Please change the units to MPa, and if you could please provide strain as %. Also please enlarge the picture for small values of stress-strain.
Line 205 Where is XRD data?
Author Response
Reviewer #2:
Comments: The reviewed manuscript concerns preparation and analysis of Graphene Oxide dispersity in PAN nanofibrous matrix. Introduction of GO filler inside nanofibers could improve several materials’ properties including mechanical one. The effect of nanofillers’ presence inside nanofibers and modifications with PDDA was studied with several methods including FT-IR, SEM, TGA and Raman. The manuscript contains many typos and mistakes but don’t need language corrections. In such form, the manuscript cannot be published.
COMMENTS TO THE AUTHOR(S):
Line 10 were instead of was>> Sorry for the mistake. We have corrected as suggested and checked thoroughly again to prevent the same mistake.
Line 19 Statistical analysis would be appreciated>> Thank you for comments. We revised as your suggestion.
These results showed that the hydrophilic and mechanical properties of GP/PAN hybrid nanofiber membranes were dramatically improved, i.e. 50% improvement for hydrophilicity and 3~4 times higher strength for mechanical property, which indicated the possibility for water treatment application.
Line 48 „Especially, GO exists…” use instead GO contain several…>> Thank you for comments. We have corrected this sentence as suggested.
Line 56 Please cite work done by Pierini et al. Titled Electrospun poly(3-hexylthiophene)/poly(ethylene oxide)/graphene oxide composite nanofibers: effects of graphene oxide reduction, where they also modified GO to increase dispersity in nanofibers.>> Thank you for comments. We have read and cited this paper as suggested.
Pierini, F.; Lanzi, M.; Nakielski, P.; Pawłowska, S.; Zembrzycki, K.; Kowalewski, T. A. Electrospun poly(3‐hexylthiophene)/poly(ethylene oxide)/graphene oxide composite nanofibers: effects of graphene oxide reduction. Polym. Adv. Technol. 2016, 27(11), 1465-1475.
Line 57-60 The sentence is so long that it is impossible to understand it.
>> Sorry for the confusion. We revised and made several short sentences for the easy understanding as below.
In this report, GO was firstly prepared from flake graphite using an improved Hummers’ method. We also modified the surface of GO with simple method in order to improve GO’s dispersibility. This modified GO was mixed with PAN polymer solution to be a homogeneous precursor solution. The final precursor solution was then fabricated as the hybrid nanofiber membranes via electrospinning process. The resulting nanofiber membranes were characterized by SEM, FT-IR, contact angle, TGA and mechanical property.
Line 77 Please remove a dot after 3 h.>> Sorry for the mistake. We have removed the dot as suggested and checked thoroughly again to prevent the same mistake.
Line 125 Figure 1. Please increase size of the pictures so that it fits the width of the text. Also, please include scales for AFM picture (X-Y) and height.>> Thank you for comments. As you suggested, we have increased the size of the pictures.
Line 131 Figure 2. Please increase the scale so that it is better visible where the peaks are. Also, for C=O and C=C there are single peaks not the range.>> Thank you for your valuable comments. As you suggested, we have increased the scale of FT-IR and we also revised the explanation of peaks as below.
Among the salient features observed are the bands at ~ 1740 cm−1, ~ 1630 cm−1 , ~ 1226 cm−1 , and ~ 1040 cm−1, corresponding to the stretching modes of C=O, -C=C, C-O, and C-O-C groups in GO respectively [32,33].
Line 151 Please do not name polymer solution a precursor solution because it is misleading.>> Thank you for comments. But polymer solutions for elelctrospinning were usually called as precursor solutions. So we only just follow this.
Line 155-156 The most important part. Since you are trying to increase the GO content in Pan fibres, and take care of its homogenous dispersion, does the analysis you perform allow you to say that it is homogenously dispersed? What about TEM analysis?>> This is a very good suggestion to prove the homogenous dispersion of GO in the nanofiber. But in this report we emphasized the homogeneous dispersion of GO not in the nanofiber but in the PAN precursor solution. However, we have tried many times to check the dispersion of GO in PAN nanofiber by TEM. As a result we could not distinguish the differences much between the TEM of GO-PAN and GP-PAN nanofibers due to the very low amount of GO in the PAN nanofibers. In this paper we have checked the blocking of needle tip by precursor solution during the electrospinning process in order to figure out the homogeneous dispersion of GO in PAN solution. What we found was that GP –PAN precursor solution did not block the needle tip while the tip was blocked continually with GO-PAN precursor solution. However, we are currently to check this phenomenon by TEM analysis and this results will be reported in the next paper.
Line 159 Since the fibres are quite large based on the SEM pictures you provide, what pore size did you measure? The pores between the fibres should be in micrometres, not in nanometres. Or maybe you mention the porosity of single nanofiber?>> Thank you for comments. We measured the avg. pore-size with a capillary porometer (Porolux 1000, IB-FT GmbH ) which is widely used to check the pore-size of membrane. Regarding the units for the pore size, we just used the units from the data of porometer. If we used micrometer for pore size, the size differences would not be pronounced, In this regard we decided nanometer was more appropriate unit to express the differences of pore size based on the GO amount of PAN solution. The measurement of pore size by porometer was reported in the other papers as below.
Jang, W.; Yun, J.; Jeon, K.; Byun, H. PVdF/graphene oxide hybrid membranes via electrospinning for water treatment applications. RSC Adv. 2015, 5, 46711-46717 Jang, W.; Yun, J.; Byun, H. Preparation of PAN Nanofiber Composite Membrane with Fe3O4 Functionalized Graphene Oxide and its Application as a Water Treatment Membrane. Membr. J. 2014, 24(2), 151-157. Line 164 The Figure 4 is useless, the info you gave in the text about darker colour of the material is enough.>> Thank you for suggestion. But we thought the evidences of color change based on the GO amount in the PAN nanofiber might be meaningful for the better understanding in terms of the homogeneous dispersion of GO in PAN nanofibers instead of just writing in the text. Of course this Figure 4 was not enough to prove the homogeneous dispersion of GO in the PAN nanofiber. Nonetheless we decided Figure 4 was necessary in this manuscript for showing the small evidence of dispersion of GO in the PAN nanofiber.
Line 167 SEM images should be increased in size, and since you used FE SEM you could provide more detail of single nanofibers.>> Thank you for comments. Firstly, we have increased the size of SEM images as suggested. In this research, the nanofiber membranes were manufactured with several layers’ nanofibers through post-treatment process so that we thought the detail of single nanofiber was not accurate due to the post-treatment of nanofiber met. However, we currently analyze the nanofiber with TEM for the detail information of nanofiber size, GO dispersion in the nanofiber, post treatment modification, and so on. These results will be reported in the next paper.
Line 188 Figure 8 Why the weight loss above 480 oC of nanofibrous material containing GO is higher comparing with PAN? It contains additionally GO, so it should be lower.>> Thank you for your valuable comment. Yes, referee is correct. However, as you see the weight loss above 480oC of GP-PAN nanofiber membrane was nearly same as the weight loss of PAN nanofiber. In fact, at high temperature we thought the PDDA made further weight loss and this led the slightly higher weight loss of GP-PAN than that of PAN. Please refer to the below TGA result showing the effect of PDDA on the weight loss from the temperature above 480oC. As a matter of fact, we are currently measuring the TGA of GP-PAN nanofiber membranes in order to prove the PDDA weight loss effect. Thank you again for your valuable information. Anyhow, this TGA information was carried out for the effect of GP on the thermal stability of PAN nanofiber membrane, and it was confirmed that GP made the enhanced thermal stability.
The effect of PDDA on the weight loss of metal electrocatalysts from Wang Shuangyin et.al., Nanotechnology, 19.26 (2008): 265601,
Line 198 Please change the units to MPa, and if you could please provide strain as %. Also please enlarge the picture for small values of stress-strain>> Thank you for suggestions. However, we just used the unit, kg/cm2, which has been known as the general unit for the mechanical strength and we think the unit change is not necessary. If you do not have any difficulty to understand the mechanical property please let me allow use this unit. Regarding the enlargement of picture we corrected as you suggested.
Line 205 Where is XRD data?>> Thank you for suggestion. This XRD data is a good idea to analyze the nanomaterial including the crystal structure. This XRD information will give us more accurate 3D structure of nanofiber membrane with reciprocal space analysis. However, in this study we tried to confirm the GO or GP in the PAN nanofiber with Raman spectroscopy and FTIR. As a result it was confirmed that GO and GP was in the PAN nanofiber. Nonetheless we think it will be better if XRD analysis was included. So we will try to measure the XRD for the next study. Thanks again for your valuable suggestion.

Reviewer 3 Report
The authors claim improved dispersibility of GP in PAN nanofibers but I could not see a sound proof of that.
The method used to measure porosity (capillary porometer under wet and dry conditions) does not seem appropriate when the hydrophilicity is changing.
Figure 5 does not provide strong evidence that pore size increases with increasing GP.
Figure 8 must show the thermal stability of all the samples.
The tensile strength is the best for the hybrid nanofiber membrane with 0.3%GP, why?
Author Response
Reviewer #3:
The authors claim improved dispersibility of GP in PAN nanofibers but I could not see a sound proof of that. The method used to measure porosity (capillary porometer under wet and dry conditions) does not seem appropriate when the hydrophilicity is changing.
>>Sorry for the unclear explanation. In this study the ‘dispersibility of GP’ meant that the homogenous dispersion of GP in the PAN polymer precursor not in the PAN nanofiber. We tried to prove this with TEM but it was not easy to recognize the differences between GP-PAN and GO-PAN due to the very low amount of GO in the nanofiber. So we decided the only evidence to show the better dispersibility of GP compare with GO was to find out the blocking problem of needle tip during the electrospinning process. In fact it was found that GO precursor solution showed the continual blocking of need tip so that the continuous electrospinning was not possible, while GP precursor solution did not occur the blocking problem. In this regard, we thought GP (GO modified with PDDA) had a high dispersibility.
>>Regarding the porosity with hydrophilicity changing, what we measured was not porosity of nanofiber membrane but pore size of nanofiber membrane by using pore size measuring equipment, i.e. porometer (Porolux 1000, IB-FT GmbH ). The hydrophilicity of surface of nanofiber membrane was proved by measuring of contact angle. The result of this was summarized in Figure 7, showing the changing of contact angle from 43o (PAN nanofiber) to 23o(GP10PAN nanofiber). This contact angle measurement has been used generally to show the hydrophilicity of the membrane surface.
COMMENTS TO THE AUTHOR(S):
Figure 5 does not provide strong evidence that pore size increases with increasing GP.>>Thank you for comments. We agree with your comment that Fig.5 does not provide the strong evidence for the increasing of pore size with increasing of GP. So we tried to make the clear evidence with the measuring values by using the pore size measuring equipment, i.e. porometer (Porolux 1000, IB-FT GmbH ). The result of this measurement was summarized in Table 2 and the average pore size of GP-PAN membrane incteased with increase of GP amount. The measurement of pore size by porometer was reported in the other papers as below.
Jang, W.; Yun, J.; Jeon, K.; Byun, H. PVdF/graphene oxide hybrid membranes via electrospinning for water treatment applications. RSC Adv. 2015, 5, 46711-46717 Jang, W.; Yun, J.; Byun, H. Preparation of PAN Nanofiber Composite Membrane with Fe3O4 Functionalized Graphene Oxide and its Application as a Water Treatment Membrane. Membr. J. 2014, 24(2), 151-157.
Figure 8 must show the thermal stability of all the samples.
>>Thank you for your suggestion. This thermal analysis was prepared only to show that the effect of GP on the thermal stability. Since we did not try to analyze the thermal properties of GP-PAN nanofiber membranes we only measured 2 samples for the comparison only. However, currently we are measuring TGA in order to figure out the effect of GP on the thermal properties, i.e. mp, Tg, and etc. These results will be appeared in the next paper.
The tensile strength is the best for the hybrid nanofiber membrane with 0.3%GP, why?>>Thank you for comments. This study aims to get the high amount of GO in the PAN nanofiber membrane by obtain the homogenous dispersion of GO in the PAN polymer precursor solution by using simple modification, i.e. PDDA treatment. As a matter fact we succeeded to get very high amount of GO in the PAN polymer precursor solution without blocking problem of needle tip during the electrospinning process. However even we prepared the high dispersion of GO the agglomeration of GO was inevitable, resulting in the discontinuous distribution. We think this phenomenon will affect the mechanical property of nanofiber membrane. In this regard the continuous distribution of GO would be a major factor in the enhancement of mechanical property, and this continuous distribution of GO was obtained with 0.3wt% of GP. Currently we are measuring TEM and hoping the result of TEM would prove our explanation. Anyhow, here is a small evidence of the effect of dispersion of GO on the mechanical property. When we used only GO for the GO-PAN nanofiber membrane 0.2wt% of GO showed the best result in terms of mechanical strength, In other word when we used GP we could get the best mechanical strength with 0.3wt% of GP due to higher dispersion of GP than GO.
Reviewer 4 Report
This manuscript reports the fabrication of modified graphene oxide/PAN hybrid nanofiber membrane via electrospinning of PAN solution containing MGO up to 1.0wt%. The obtained membrane show improved hydrophilic and mechanical properties. However, it is noticed that the authors published a very similar paper titled “Highly Controlled Integration of Graphene Oxide into PAN Nanofiber Membranes” in Appl. Sci. (Appl. Sci. 2019, 9, 962; doi:10.3390/app9050962) recently, making the reviewer very hard to identify the new significant contribution of this manuscript. It seems there is only some slight difference between these two manuscripts. Thus, the manuscript is not recommended to be published without demonstrating obvious new progress.
Author Response
Reviewer #4:
This manuscript reports the fabrication of modified graphene oxide/PAN hybrid nanofiber membrane via electrospinning of PAN solution containing MGO up to 1.0wt%. The obtained membrane show improved hydrophilic and mechanical properties.
COMMENTS TO THE AUTHOR(S):
However, it is noticed that the authors published a very similar paper titled “Highly Controlled Integration of Graphene Oxide into PAN Nanofiber Membranes” in Appl. Sci. (Appl. Sci. 2019, 9, 962; doi:10.3390/app9050962) recently, making the reviewer very hard to identify the new significant contribution of this manuscript. It seems there is only some slight difference between these two manuscripts. Thus, the manuscript is not recommended to be published without demonstrating obvious new progress.>>Thank you for your valuable comments. Sorry to make you confuse. However, this paper has the pronounced difference compare with the former paper published in Appl. Sci. Of course the main difference was the modification method of GO. This paper used PDDA while the former paper used cetyltrimethylammonium chloride (CTAC). As you see these two different modifications showed different amount of GO in the PAN polymer precursor solution. PDDA made 0.1~1.0wt% of GO while CTAC made 4~30wt% of GO. Even though CTAC made very high content of GO in the PAN nanofiber membranes the mechanical property was much lower than GP-PAN nanofiber membranes. This indicates the modification of GO would affect much in terms of mechanical property. In adding to that the purpose of those two papers was totally different. This paper aims to find out the effect of GP on the properties of GP-PAN nanofiber membrane, i.e. mechanical property, surface property, thermal property and pore property. In adding to that we tried to figure out the optimized GP wt% for the best property. On the other hand the former paper aimed only the surface modification method to get high content of mGO (GO modified with CTAC). Surprisingly we could get the more than 30wt% of mGO in the PAN polymer precursor solution without the blocking problem of needle tip during the electrospinning process. But we failed to get more than 1.0wt% of GP in the PAN polymer precursor solution. With this paper we found that the dispersion of GO in the PAN nanofiber membrane played an important role in determining of properties of nanofiber membrane. Currently we are carrying out the TEM experiments with XRD measurement, and hoping to get clear explanation in terms of the effect of GO dispersion. These results will be reported soon.
Round 2
Reviewer 2 Report
The authors corrected the manuscript accordingly to the previous comments.
Author Response
Fabrication and characterization of modified graphene oxide/PAN hybrid nanofiber membrane
Response to comments by Reviewers:
We greatly thank all Reviewers for their constructive feedback of our work and for the helpful comments.
Reviewer #2:
Comments: The authors corrected the manuscript accordingly to the previous comments
>> Thank you very much for your spending time of manuscript reviewing. We appreciate your valuable comments and suggestions regarding our manuscript

Reviewer 3 Report
The revised version fails to provide strong proof of the authors claims. It was disappointing that the required data will be part of a new manuscript, regardless that they are critical for the present one.
Author Response
Fabrication and characterization of modified graphene oxide/PAN hybrid nanofiber membrane
Response to comments by Reviewers:
We greatly thank all Reviewers for their constructive feedback of our work and for the helpful comments.
Reviewer #3:
Comments: The revised version fails to provide strong proof of the authors claims. It was disappointing that the required data will be part of a new manuscript, regardless that they are critical for the present one.
>> Thank you for your valuable comments. We tried to our best to prove the originality, logical development, and excellent academic value of manuscript. But if you feel the revised version fails to provide strong proof of the Authors, this will be our responsibility. However, we will keep in our mind your comments, and this will be a great help for us to prepare another manuscript.

Reviewer 4 Report
The authors explained the difference between this work and previous work, which is quite convincing. It can be accepted for publication after checking the text carefully to avoid any typos.
Author Response
Fabrication and characterization of modified graphene oxide/PAN hybrid nanofiber membrane
Response to comments by Reviewers:
We greatly thank all Reviewers for their constructive feedback of our work and for the helpful comments.
Reviewer #4:
Comments: The authors explained the difference between this work and previous work, which is quite convincing. It can be accepted for publication after checking the text carefully to avoid any typos.
>>Thank you very much for your spending time to review our manuscript including your valuable comments. As your suggestion we checked the text again very carefully and found several typos.

Round 3
Reviewer 3 Report
TGA (Figure 8) should always include all the components studied.
The claim that PAN/GP hybrid nanofiber membraness could be a promising candidate for high-performance secondary battery separators is too strong and requires some electrical or electrochemical characterization.
